# Bio-Interactive Zwitterionic Dental Biomaterials for Improving Biofilm Resistance: Characteristics and Applications

**DOI:** 10.3390/ijms21239087

**Published:** 2020-11-29

**Authors:** Utkarsh Mangal, Jae-Sung Kwon, Sung-Hwan Choi

**Affiliations:** 1Department of Orthodontics, Institute of Craniofacial Deformity, Yonsei University College of Dentistry, 50-1 Yonsei-ro, Seodaemun-gu, Seoul 03722, Korea; utkmangal@yuhs.ac; 2Department and Research Institute of Dental Biomaterials and Bioengineering, Yonsei University College of Dentistry, Seoul 03722, Korea; jkwon@yuhs.ac; 3BK21 FOUR Project, Yonsei University College of Dentistry, Seoul 03722, Korea

**Keywords:** zwitterionic polymers, biofilm, anti-biofouling, carboxybetaine methacrylate, sulfobetaine methacrylate, 2-methacryloyloxyethyl phosphorylcholine

## Abstract

Biofilms are formed on surfaces inside the oral cavity covered by the acquired pellicle and develop into a complex, dynamic, microbial environment. Oral biofilm is a causative factor of dental and periodontal diseases. Accordingly, novel materials that can resist biofilm formation have attracted significant attention. Zwitterionic polymers (ZPs) have unique features that resist protein adhesion and prevent biofilm formation while maintaining biocompatibility. Recent literature has reflected a rapid increase in the application of ZPs as coatings and additives with promising outcomes. In this review, we briefly introduce ZPs and their mechanism of antifouling action, properties of human oral biofilms, and present trends in anti-biofouling, zwitterionic, dental materials. Furthermore, we highlight the existing challenges in the standardization of biofilm research and the future of antifouling, zwitterated, dental materials.

## 1. Introduction

Dental biomaterials form the backbone, which helps prevent, restore, and rehabilitate oral form and function. Designing and testing these materials produces various mechanical and biological properties, leading to their harmonious presence inside the oral cavity. The last few decades have seen notable developments in dental materials, and properties such as adhesion to the tooth surface and biomimetics have been extensively explored [1]. All of the approved materials meet the biocompatibility standards, but they do not account for the interfacial interaction with oral microcosms [2,3].

Methacrylate-based resin composites and polymers are the main components of dental restorative materials used to address the restoration of carious and missing teeth [4]. These materials meet most of the standard requirements but continue to be afflicted by issues concerning premature mechanical and esthetic failures [5]. These failures are multifactorial, but interaction with the host-factors is considered as a key factor [6,7]. In the host-factors, the critical factor of microbiological interaction at the surface leads to biofilm formation [8]. The etiologic role of oral biofilms in dental caries and periodontal diseases has been previously documented [9]. The improved scientific evidence also indicates an active interaction at the dental materials and oral biofilm interface. The majority of the studies, including clinical trials concerning biofilm research, are related to stomatognathic diseases and address a wide range of issues such as caries, periodontitis, and demineralization [10]. Nevertheless, it is essential to note that the interfacial interaction between the dental material and biofilm is multifactorial. Subsequently, research on a variety of methods are focused on improving this interaction.

Dental biomaterials differ from the hard tissues in the oral cavity in terms of both their physical and chemical properties. Advancements in biomimetic restorative materials are rapidly occurring, but they exhibit a different surface interaction. The key surface properties of topography, roughness, surface energy, and the inherent chemical composition alter the biofilm interaction at the surface of the restorative materials [11,12,13]. Materials with low surface energies are associated with a higher affinity for bacterial adhesion, and many dental materials present higher surface energies than natural enamel [14,15]. Dental plaque is an ecologically dominant form of oral biofilm associated with cariogenic and periodontal infections. The increase in surface microroughness and topography is correlated to the surface energy and plays a defining role in microbial adhesion. A rough surface leads to an increase in the difficulty of removing dental plaque like biofilms with conventional methods (e.g., mechanical brushing) alone. Microroughness, expressed as the arithmetic mean (Ra) value, has been suggested for comparing the surface roughness of the materials, and a maximum threshold of 0.2 μm has been proposed to limit microbial adhesion [14]. Biofilm formation showed considerable variation, independent of the bacterial adhesion correlation to the prescribed threshold [16]. Hence, regardless of the surface finish, dental materials lack any inherent biofilm resistance and are prone to secondary infections.

Broadly, two approaches have been reported for the management of oral biofilm formation, comprised physical and chemical methods. While physical methods involve mechanical removal of the biofilm by actions such as tooth brushing, conventional chemical methods include dentifrices [17] and dental mouth rinses [18]. These approaches have focused on substituting or adding chemicals and have displayed good efficacy in an in vitro environment. However, these approaches have inherent limitations, such as dependence on user compliance, and a tendency to damage restorations (e.g., by esthetic and strength deterioration) and oral tissue (e.g., due to hypersensitivity, desiccation, and discoloration) [19].

Apart from adhesive resins, the methacrylate-based resins used in the fabrication of prosthodontic, removable appliances are also limited by issues of plaque biofilm formation [20]. Plaque biofilm formation on acrylic-based appliances is the cause of secondary fungal and bacterial infections, leads to a degradation of the polymer constituents, and reduces the appliances’ longevity [21]. The challenges mentioned above have laid the foundation for the development of novel approaches to counter biofilms. Many of these new methodologies, currently under research, involve modification of the dental materials by using antibiotics, incorporation of metal oxides [22], nanoparticles [23], and anti-adhesion coatings using hydrophilic polymers [24].

Amidst a large number of methods, the approach utilizing the resistance to biofilm formation due to anti-adhesion properties has attracted attention. This property is characteristically reported for zwitterionic polymers (ZPs). The innovation and application of ZPs is a close form of biomimicry against fouling, inspired by the mammalian cell phosphatidylcholine membrane, which is characteristically present on the outer surface [25,26]. These polymers are formed by an equal amount of anionic and cationic groups on their chains with a high dipole moment. With a net charge of zero, these polymers present superior surface lubrication, antifouling, and biocompatibility. This leads to a wide range of multi-disciplinary adaptations of these polymers, including dental biopolymers [27].

Therefore, it is imperative to provide an overview of current progress on the developing role of ZPs in dental biomaterial science. In this scoping review, ZPs are briefly introduced, followed by oral biofilm characterization, present trends in the research, and the development of zwitterionic dental materials reported in recent years. In the last section, the existing challenges and road to the future development of oral biofilm-resistant materials are highlighted. We hope that this review will stimulate more innovative ideas to advance research and point toward the persistent question of “what next?”.

## 2. Theoretical Aspects of Zwitterionic Polymers

The use of hydrophilic polymers such as poly (vinyl alcohol) and poly(ethylene glycol) (PEG) has been suggested to address the problem of surface fouling. PEG has been used as the gold standard of nonfouling polymers as a biocompatible polymer but has poor stability due to autoxidation [28]. The degradation of the PEG chains further increases with an increase in temperature above 35 °C and an ionic environment, such as hemodynamics [29]. ZPs form the closest alternative to PEG when considering amphiphilicity. ZPs containing an equal number of cations and anions have been classified based on their chemical structure.

ZPs with anion and cation groups existing on the same monomeric units form polybetaines (also known as polyzwitterions), such as poly (sulfobetaine methacrylate) (SBMA), poly (carboxybetaine methacrylate) (CBMA), and poly (2-methacryloyloxyethyl phosphorylcholine) (MPC). When both ion groups are present on different monomers, the ZPs are of the polyampholyte type, such as 2-(dimethylamino) ethyl methacrylate and methyl methacrylate block copolymer (DMAEMA-MAA). All ZPs can exist in different architectures, among which the membrane and brush type are the most commonly grafted types.

The distinctive feature of ZPs occurs due to its antipolyelectrolyte effect (APE) [30,31] and super-hydration ability [24]. The salt-responsive state of ZPs, which also affects their viscosity in a solution with low-molecular salt (LMS), is regarded as the opposite of the poly-electrolyte effect, hence it is termed as the APE. This property, detailed by Georgiev et al. [30], has been extensively explored. Accordingly, the introduction of LMS is believed to reduce the interchain dipole interaction within the polymer. The change in the polymer’s charge interaction promotes the extension of the chains, and subsequent swelling of the globularized polymer occurs. The reduction in the interchain interaction encourages the polymer-water interaction to achieve an entropy balance, resulting in the formation of a tightly-bound water layer [32]. This characteristic, bound, water layer formation is the second unique feature of ZPs, referred to as their super-hydration ability (Figure 1).

The super-hydration ability, characterized by the formation of the hydration shell via electrostatic interaction, is also influenced by the arrangement of water molecules within the shell. The arrangement mimicking free water causes an increase in the affinity to water [33]. Therefore, the formation of a compactly-bound, thick, energetic layer occurs on the ZP brush. This layer enforces a strong steric effect, facilitating a strong resistance to protein adsorption and subsequently to fouling [26]. In other words, the development of biofilm resistance takes place as a combination of surface hydration and steric repulsion [34].

## 3. Human Oral Biofilm: Composition and Properties

The microbiome has been described as the presence of pathogenic, symbiotic, and commensal microorganisms together in a community, thus, an oral biofilm can be considered a sub-microbiome [35]. Moreover, the biofilm is a microbially-derived, sessile community wherein cells that are irreversibly attached to the substrate or each other are embedded in an extracellular polymer substance (EPS) matrix [36]. Advancements in genomic studies have helped decode the oral microbiome, evidencing a broad heterogeneous nature of the oral microbiota with more than 600 different species coexisting [37,38].

The development of oral biofilms is a spatiotemporal phenomenon that is composed of initial, physicochemical interactions. This initial contact establishes the foundation for the biofilm’s growth and maturation, incorporating a nutrient, pH, and oxygen level gradient [39]. The multi-stage biofilm life cycle is described in detail in the review by Stoodley et al. [40], and the biofilm has also been recently discussed as an “emergent form of bacterial life” by Flemming et al. [41]. Therefore, to better understand the mechanisms that help combat biofilm formation in the oral cavity, it is important to understand the development of this emergent life form.

The formation of the biogenic habitat begins when free-floating bacteria adhere to the surface. This attachment of pioneer surface colonizers is mediated by the salivary components that attach to the tooth surface and form a pellicle [42]. The pellicle alters the surface chemistry, including the surface charge and energy, commencing initial reversible and then irreversible surface adhesion of the microbes. The nature of the adhesion is mediated by the interaction forces, which can be van der Waals forces, electrostatic interactions, or specific adhesion to the surface. Following irreversible attachment to the surface, the bacteria begin to multiply rapidly.

Along with active replication, EPS produce glycocalyx [43]. Biofilm maturation is regulated by prokaryotic gene expression and cell-to-cell signaling (quorum sensing (QS)). In addition, the availability of nutrients, hydrodynamic flow, pH, pO_2_, osmolarity, and elimination of toxic metabolites affect the biofilm [44]. The last stage of the biofilm cycle occurs when the film reaches a critical point of equilibrium such that the outermost surface begins to release free bacteria and EPS globules. These organisms escape the habitat to colonize at a new surface or join another maturing biogenic habitat (Figure 2).

The hydrodynamic flow in the biofilm creates a gradient of resources such as nutrients, pH, and oxygen saturation. These gradients give rise to phenotypic and genotypic heterogeneities within the biofilm [41] (Figure 3). Heterogeneities lead to coexisting microbes lying on a spectrum, including active pathogenic, persister, and viable but not culturable (VBNC) types of cells. Consequently, this increases the challenge of managing biofilm growth by antimicrobials alone.

Due to the complexity of the biofilm, Lin [35] proposed a categorization of the four attributes of the biofilm. First, the constituents, which describe the materials that comprise the biofilm, including the microorganisms, viruses and particles, extracellular matrix, signaling molecules and enzymes, and debris. Second, the quantity of the biofilm, referring to the overall biomass, number, and concentration of the heterogeneous biofilm. In addition, the amount and constituents of EPS are also included in this category. The third attribute is the biofilm structure, which is related to the arrangement of the constituents. It is correlated with both the QS and the resistance of the film to shear forces. Lastly, biofilm function characterizes the pathogenicity of the biofilm, which also entails the metabolic activity, gene expression, and mechanism of surface adhesion. Each of the above attributes is associated with a measurand, which can help to analyze the efficacy of the intervention being used to prevent/remove the biofilm, as shown in Figure 4.

## 4. Biofilm-Resistant Dental Materials

In recent years, the dynamic nature of biofilm interaction with dental materials has been widely recognized. The increased focus on the longevity of dental materials and the challenge of secondary caries and other recalcitrant disease forms [45] have driven biomaterial research into the development of robust materials. While biofilm characteristics occupy a large area of the research focus, it is essential to state here that the underlying substrate acts as a significant factor in defining these characteristics [35]. In the following sections, we review the literature discussing different dental materials based on their clinical applications. In addition, the review will focus more on polybetaine ZPs, with or without polyampholytes. Although more than 40% of recent research has included cationic quaternary ammonium monomers (QAM) as additives for biofilm resistance [46], this topic has been reviewed in detail by Makvandi et al. [47].

### 4.1. Restorative Materials

Resin-based materials are one of the most common types of dental materials used intra-orally. Applications of resins include luting agents, base cement, and bulk restorative cement. In the research reported by Zhang et al. [48], an initial attempt was made to benefit from the protein-repellent action of MPC. They combined QAM (dimethylaminohexadecyl methacrylate (DMAHDM)) with MPC to develop a resin with a synergistic effect. Their study aimed to combine the antibacterial action of QAM with the protein-repellent characteristics of ZPs, represented by MPC in this case. The results from this study revealed two notable findings: the possibility of synergism between the two additives, and the independent potency of ZPs to resist protein adsorption and bacterial fraction. This result was expressed with MPC at 3 wt.% and with QAM at 1.5 wt.%. Cherchali et al. [49], with a similar aim, explored the synergism of MPC with QAM and dimethyl-hexadecyl-methacryloxyethyl-ammonium iodide (DHMAI). Both studies had an organic phase of 30%, but Cherchale et al. compared the MPC concentration at 7.5% and 10% of the resin phase. In addition, they compared the formulations against three existing commercial resins. However, they rejected the hypothesis of synergism because no added effect was elicited from MPC. This was rationalized by emphasizing that the low concentration of MPC included in their study was limited due to an observed, significant reduction in the mechanical properties.

The synergistic action of DMAHDM with MPC was further explored by the same lab in a calcium phosphorous-containing, bioactive nanocomposite at the same mass fraction of 3% [50]. This nanocomposite showed the potential for maintaining the physical property of strength while being effective against biofilm formation and bacterial activity. They also showed that the pH was improved compared to the commercial control in their experiment, notably above the 5.5 mark of the critical pH. This work was further extended to evaluate the same nanocomposite without DMAHDM and using two different mass fractions of MPC at 1.5% and 3% [51]. This study concluded that a 3% mass fraction of MPC was most efficacious against biofilm formation and had no effect on the bioactive nature of the nanocomposite.

The influence of direct incorporation of MPC in a resin-based composite (RBC) with a surface pre-reacted, glass-ionomer filler was investigated in detail for different mass fractions of MPC [52]. The MPC was mixed in a powder at 1.5, 3.0, 5.0, 7.5, and 10.0 wt.% with the RBC and was evaluated against the unmodified RBC as the control group. Similar to the earlier study, a trend showing a reduction in the physical and mechanical properties with an increase in MPC concentration was observed in the results of this study. However, the changes in pH and the ionic concentration observed in this study were of interest. MPC at 5 wt.% showed a rapid increase in pH compared to the control groups and lower percentages. However, the effect did not increase beyond 5 wt.%. With respect to ion release, the results indicated an increasing trend of sodium, phosphate, and fluoride release, with an increase in the amount of MPC added. According to the authors, this effect was initiated with the addition of 1.5 wt.% of MPC and was similar up to 3 wt.% but showed a significant increase in ionic release above 5 wt.%, especially for phosphate ions. The phosphate ion concentration changes were most noteworthy, with more than a nine times increase from 1.5 to 10 wt.% of MPC. In the test for bacterial adhesion against *Streptococcus mutans* (ATCC 25175) and *Actinomyces naeslundii* (initial colonizer), they reported a significant reduction in mass fractions of 1.5 to 7.5 wt.% with the lowest numbers in the 3 wt.% group. The same concentration also exhibited significant resistance against *Veillonella parvula* (early colonizer) and *Porphyromonas gingivalis* (late colonizer). Subsequently, the 3 wt.% group also showed significant resistance to biofilm formation with a lower biomass and thickness (Figure 5). Multispecies biofilm resistance of the MPC was again confirmed in another study, which evaluated biofouling resistance against an increasing number of bacterial species in the biofilm constituents (1,3,6, and 9) and compared the biofilm metabolic activity and cellular quantity [53].

### 4.2. Varnish and Sealants

The use of preventive measures for combating dental caries is one of the leading aspects of oral biomaterial research. The research focuses on varnishes and sealants to form an impermeable, mechanical barrier, however, the resin sealants are not inherently immune to biofilm formation.

Taking this into account, Kwon et al. [54] investigated the influence of MPC as an additive in light-curable fluoride varnish (LV). They experimented with the addition of MPC at a wide range of mass fractions at 1.5, 3.0, 5.0, 10.0, 20.0, and 40.0 wt.%. The addition of MPC at high percentages showed a tendency for an increase in the film thickness, with significantly increased values above 5 wt.%. The degree of conversion, protein adsorption, and *S. mutans* attachment also showed a similar pattern, with desirable effects limited to a maximum of 5 wt.%. Although significant effects were observed up to 5 wt.%, due to the absence of any marked difference with the 3 wt.% group, they recommended the addition of 3 wt.% for future use.

Expanding the research focus, a follow-up study [55] investigated the incorporation of ZPs, in the form of CBMA, MPC, and SBMA, into the LV. Using a mass fraction of 3 %, as established in the above studies, they compared the three polybetaines for their influence on biofilm accumulation. Resistance to protein adsorption was observed in all ZP groups, and subsequently, a lower adhesion of bacteria and saliva-derived biofilm mass and thickness were also reported. In addition, this study also showed protective action against demineralization, which can be considered as a result of reduced biofilm formation (Figure 6).

Investigations on resin-based sealants incorporating ZPs also mirrored many similar findings. The addition of SBMA into the sealant was investigated for 1.5, 3, and 5 wt.% [56]. In the commercial, resin-based, sealant incorporation of SBMA, up to 3 wt.% showed significant resistance to the attachment of *S. mutans* with no increase in cytotoxicity. However, this study was limited to a single bacterial species attachment and did not evaluate the biofilm formation response.

A recent study on sealants investigated potential synergism by the addition of MPC and bioactive glass (BAG), compared to a commercial control [57]. In addition to the common findings seen in previous resin-based studies, this study also tested the impact of aging on the biofilm resistance of the ZP-incorporated sealant. Commercial and experimental control groups and 3 wt.% MPC containing the sealant group were compared with a group containing 12 wt.% BAG + 3 wt.% MPC. Similar to the restorative resin study, a significant large spike in the phosphate ion release was reported in the combination group, which was again attributed to the action of MPC. However, this was not experimentally investigated.

The protein adsorption investigation was similar for MPC-containing groups. The biofilm resistance was evaluated after aging in static immersion, and four weeks of simulated thermocycling showed potential for the long-term anti-biofouling effect of MPC in the resin-based composites (Figure 7). It is believed that thermocycling and static immersion impacted the resin structure and thereby the surface chemistry, leading to an increase in the tendency for bacterial adhesion. Hence, the results of this study provided insight into the significance of MPC as an additive in preventive resin development in the future.

### 4.3. Adhesive

Restoration and attachment of the tooth involves bonds using a system of adhesives and primers. These adhesives become the microscopic interface between the tooth and the restorations and, if the adhesive agent is prone to degradation or loss, can lead to microleakage and subsequent secondary ailments. This dissolution of the adhesive is primarily attributed to the localized acidic environment created by plaque biofilms [58,59]. This technique-sensitive step in restorations has long interested researchers in developing an adhesive resistant to dissolution under a microbial-rich environment.

An adhesive with MPC incorporated in the dentin bonding agent was investigated at 3.75, 7.5, 11.25, and 15 wt.%, with an aim to resist biofilm formation and consequent damage of the adhesive structure [60]. The 7.5 % MPC mass fraction dissolved in the primer solution of the commercial adhesive was most effective in resisting protein adhesion and *S. mutans* bacterial attachment. Although all percentages of MPC were effective in preventing protein adsorption, at 7.5 wt.%, the dentin shear bond strength could be preserved most effectively. Research regarding the further advancement and combination of MPC with DMAHDM in the adhesive was also analyzed [61]. The detailed analysis of the biofilm with this combination validated the synergy concept, similar to the restorative composite. An augmented effect on the reduction of bacterial number, viability, metabolic activity, and biofilm extent was observed.

The use of MPC to augment pH neutralization of the dental adhesive was also reported in an in vitro setup [62]. This study tested the MPC concentrations of 2.5, 5, and 10 wt.%, as the amine monomer component of the resin adhesive, for assessment of the neutralization capacity and kinetics. It was designed based on the rationale that the crosslink density varies with the type of amine monomer [63]. This particular study focused on the physical properties of MPC-containing, dental resin adhesives, but provided an important insight into the relation of MPC with the resin. The most significant observation was that the neutralization rate was enhanced with the addition of MPC as a monomeric, amine-containing formulation. Thus, along with biofilm resistance, the addition of MPC offers an increase in pH, aiding in the neutralization of the milieu around resins.

Orthodontic adhesives used in the direct attachment of the brackets to the enamel are also highly prone to plaque accumulation [64]. This leads to damage to the surrounding enamel structure, which manifests as white spot lesions [65]. Although multifactorial, a key role of the adhesive has also been reported with resin-modified glass-ionomer cement (RMGIC), with a high tendency for plaque accumulation [66]. While mechanical means are effective in eliminating plaque build-up, modification of the adhesive cement forms a compliance-free alternative. To this end, MPC has been incorporated into RMGIC [67] and orthodontic bonding, flowable, resin-based agents [68]. RMGIC was tested by supplementing commercial control with 1.5 wt.% DMAHDM and 3 wt.% MPC and showed a predictable reduction in protein and bacterial adhesion. The flowable resin adhesive evaluated the synergistic effect of mesoporous, bioactive glass with 3 wt.% MPC and reported an improvement in anti-demineralization, in addition to the antibacterial and protein repellant action. Although MPC presented a reduction in bacterial and protein adhesion, it is important to consider that the inherently different materials will interact with the biofilm differently. RMGIC will show surface degradation under aciduric, bacterial biofilms [69], and toxic bacterial metabolites, such as esterases, will lyse and leach out resins [70].

### 4.4. Endodontic Materials

The anatomical and morphological intricacy of the root canal system in human teeth has led to a diverse and complex endodontic biofilm milieu [19]. Biofilm by the region of formation can be intracanal, extra-radicular, or on the surface of the endodontic biomaterial [71]. Many studies have evaluated residual bacteria in the root canal system after a different stage of root canal therapy [72]. These residual bacteria develop a new biofilm microenvironment and lead to treatment failure and recalcitrant infections [73]. Thus, researchers have focused on several modifications to improve the efficacy of irrigants and cement [74]. These intracanal materials should ideally be effective against the microbiota while being compatible with local and systemic circulation [75].

To this end, using ZPs, efforts have been made to enhance cement and sealers with MPC. Calcium silicate-based cement was modified by the incorporation of MPC and investigated in detail for 1.5, 3.0, 5.0, 7.5, and 10% by mass fraction [76]. The MPC was mixed into the commercially available mineral trioxide aggregate. This study also evaluated the mineralization ability of the cement in addition to the known characteristics of protein adsorption and bactericidal effects. Both 7.5 and 10 wt.% addition showed a depreciation in ease of manipulation and the basic physical characteristics of strength and wettability. The cement performed best when 3 wt.% MPC was included, showing the highest resistance to adhesion of *Enterococcus faecalis*. However, the bactericidal effects of the cement were not elicited for any concentration of MPC in the cement. Further testing with 3 wt.% MPC also showed an improved calcium deposition in the alizarin red staining results (Figure 8).

In another study, an investigation was conducted on root canal sealer with pyromellitic dianhydride glycerol dimethacrylate, calcium phosphate, DMAHDM, and MPC in various combinations [77]. The study experimentally verified bacterial resistance against the abundant group of endodontic strains [78]: *E faecalis, A naeslundii,* and *Fusobacterium nucleatum*. A significant reduction was observed in the colony-forming unit of all bacterial strains with groups containing MPC. However, in a group containing MPC with DMAHDM, a comparative reduction in the release of calcium and phosphate ions was observed in the results.

### 4.5. Removable Appliances

In dental healthcare practice, poly (methyl methacrylate) (PMMA)-based acrylic resins are used for the fabrication of removable appliances and maxillofacial prostheses. Most of these appliances are intended for long-term use. However, acrylic resin appliances present a high risk of bacterial attachment and dental plaque formation [79,80]. The proteome of the acquired pellicle on the acrylic surface modulates the virulence of the biofilm by influencing the cell function of the primary colonizers by the cellular signaling mechanism [81]. This has been particularly observed for *C. albicans* biofilms, where increased enzymatic activity causing localized immunosuppression was also observed [82]. The risk of severe nosocomial infections and resistance to drug therapy makes it essential to develop methods to prevent the growth and spread of such virulent microbiota.

The deposition of plaque biofilms on the surfaces of dentures has been a troublesome factor that directly affects the local and systemic health of the patients [83]. Management strategies to prevent plaque biofilm formation on the acrylic surface of removable appliances have been attempted using two primary methods: (1) adding microbial-resistant filler components at the micro and nanoscale [23] and (2) improving the surface properties of the fabricated appliances by coatings [84,85]. The ZPs show the possibility of action via both of the above interventions, and multiple studies in recent years have detailed promising findings in this direction of work.

In comparison to MPC, CBMA has been reported to exhibit higher differences in charge density, showing low self-association and hydrophilicity [86,87,88]. This effect was recently explored by incorporating CBMA in auto-polymerizing PMMA for dental applications [89]. As MPC has been widely researched, it was considered as the positive control, while the different concentrations of CBMA acted as the experimental groups. Five different concentrations of CBMA (0.75, 1.5, 3, 5, and 7.5%) were mixed into the monomer liquid component to make the samples. The superior hydrophilicity of CBMA was evidenced in the characterization of the samples by Raman spectroscopy. Furthermore, CBMA at 3 wt.% showed the most acceptable balance between mechanical properties and bacterial resistance. Although the protein adsorption resistance was similar to that of MPC, the computed human saliva biofilm mass was significantly lower than that of 3 wt.% of MPC. These results were concordant in both the immediate and aged samples (Figure 9).

In addition, CBMA showed superior resistance to *C. albicans* with a greater than 3-log fold reduction in adherence, compared to the control group. Similar trends were also observed against *A. naeslundii* and *V. parvula* adherence on the 3 wt.%, CBMA-incorporated, PMMA disks (Figure 10).

In addition to direct incorporation into the resin matrix, MPC coating and grafting have also been investigated for their efficacy in preventing biofilm formation on dentures. Heat-polymerized PMMA resin was surface-modified using two methods [90]. The first method involved adsorption of radically copolymerized MPC and butyl methacrylate (BMA), and the second involved the surface-initiated direct grafting of MPC. The grafted MPC displayed a significantly higher hydrophilicity and improved resistance to protein and *S. mutans* biofilm formation. The surface-grafted MPC PMMA also showed a better tolerance to toothbrush abrasion cycling.

In a similar study, MPC was copolymerized with BMA to prepare a photoreactive polymer [91]. This modification was conducted to overcome the challenges of a durable coating of MPC on the PMMA surface. The in vitro analysis showed that the MPC coating was formed on the PMMA surface and was resistant to *S. mutans* biofilm formation. The in vivo follow-up of this project was reported in a recent clinical study [92]. This study compared the effects over a two-week interval by calculating the plaque index and quantifying the optical density of the constituent biofilm bacteria. A promising improvement in both the measurands was reported with the surface treatment of an MPC-containing photopolymer (PMBPAs).

Although the positive impact of the MPC against salivary and single-species biofilms has been validated in multiple studies, detailed analysis of specific strains of the species has not been widely reported. A recent study evaluated *C. albicans* (16 strains), Non-*C. albicans Candida* (NCAC; 10 strains) and Methicillin-resistant *Staphylococcus aureus* (MRSA; six strains) resistance of the MPC-polymer-coated, acrylic, denture plates [93]. The microbial adherence was tested on the mucin-coated specimens by comparing the cell hydrophobicity. All strains tested showed a generalized reduction in microbial adhesion, with the highest reduction shown for the 5% MPC-polymer. Although concentration-dependent results showed a significant reduction, there was no influence of the MPC on the cellular growth for all of the strains of fungi and bacteria evaluated.

A method to improve MPC grafting and experimentally validate the surface hydration layer was conducted by grafting MPC onto an orthodontic retainer [24]. By modulating the grafting conditions between the deionized MPC solution and the 2.5 M NaCl solution, the hydration bond strength, bacterial adherence (*A. naeslundii.*, *S. aureus.*, and *P. aeruginosa*), and salivary biofilm resistance were determined. Both models showed an improved performance in comparison to the control specimen. However, the NaCl ionic model showed a significant reduction in adherence to *A. naeslundii* and salivary biofilm mass deposition.

While the antifouling properties of MPC have been widely explored, research to develop substitutes for non-phosphatidylcholines-based, polymeric coatings on acrylic, denture surfaces has been developing simultaneously. Sulfobetaine-based, polymer application on PMMA has shown resistance to *P. aeruginosa, S. epidermis, S. aureus,* and *C. albicans* [94]. Coating with a sulfobetaine analog has attracted interest because it can be synthesized in a controlled manner, maintaining the copolymer architecture, and is a relatively cost-effective method.

Atomic transfer grafting of SBMA on the PMMA surface has shown an excellent biofouling resistance to bacterial adhesion and biofilm formation, suggesting the longer chains and higher densities of nonfouling groups [95]. A similar, potent, antifouling effect was also observed in the photopolymerized coating of SBMA in comparison to other hydrophilic polymers, however, the results were expressed only for a short-term effect [96]. Recently, using the sulfobetaine group in SBMA, a cross-linkable copolymer containing sulfobetaine methacrylamide coatings on the denture base resin were investigated for resistance to *C. albicans*. The modification was performed by swapping the ester group with the amide group to improve the hydrophilicity and stability, and the XTT-reduction assay results validated an effective micro-coating [97].

### 4.6. Hydroxyapatite, Surgical Membranes, and Implants

Bone grafts and implants are biomaterials used in the rehabilitative treatment of dentoalveolar apparatus. However, the clinical success rate shows marked variations which has also been attributed to the adhesion of the oral microbes. The formation of biofilm can impair the local healing response and result into treatment failure [98,99]. Considering the significance of implant surface properties, multiple efforts have been made towards modifying implant surfaces by application of surface coatings [100], plasma treatment [101], or by reducing surface roughness [102]. However, both graft and implant materials depend on additional surface modifications, thus warranting an enhancement of material properties to resist microbial colonization.

Hydroxyapatite (HA) is a bioceramic with a base of calcium and phosphate and is widely used in reconstructive surgery. Surface functionalization of HA with ZPs has been explored in surface-functionalized, nanocrystalline HA [103]. Functionalization of the surface of HA, both as powder and an additive-manufactured macroporous scaffold with ZPs, showed improved resistance to bacterial adhesion against *E. coli* while maintaining biocompatibility and osteoblastic promotive action.

Another effort toward the modification of HA was made by coating it with MPC-polymer-containing, calcium-binding moieties, 2-methacryloyloxyethyl phosphate (MOEP) [33]. This study investigated the resistance to protein adsorption, cell adhesion, and *S. mutans.* The results of the study indicated the most effective response occurred from the 50% MPC-containing copolymer with optimized calcium ion binding. The biomimetic HA structure to the enamel surface thus suggests a larger possibility for the application of such resistant bioceramics.

Alveolar bone defect treatment uses a barrier membrane (GBR) along with bone grafting to create a secluded space. The primary function of this membrane is to permit selective growth of osteoblasts on the graft side while preventing the invasive proliferation of rapidly multiplying fibroblast cells [104]. The surface hydration and energy characteristics of the ZPs resist cellular attachment, and the application of this feature aids in preventing fibroblastic invasive growth. This was experimentally verified in a recent study with an MPC-based polymer (poly (MPC-co-TSMA) (PMT)) layer on the outer surface of the GBR membrane [105]. The ZP coating was applied in relation to Janus chitin nanofibers. While the chitin side of the membrane promoted osteoblast development, the MPC-coated side showed good resistance to epithelial migration.

## 5. Challenges and Future of Oral Biofilm Research

A notable variation can be seen in the scientific literature when analyzing oral biofilms. Independent of the analytical method used in the evaluation, it is important to understand the limitations of the technique used, the measurand in question, and, subsequently, the experimental control and interpretation of the findings.

The need for standardized methodology in dental research also extends to the investigation of the impact of biomaterials. In biofilm research, the establishment of the “Minimum Information about a biofilm experiment (MIABiE)” guidelines in 2014 emphasized the communication of metadata and consistent vocabulary for biofilm-related results [106]. In addition, regarding reproducibility, the applicability of the simulation model is also critical in defining clinical relevance [6]. The initial screening of the efficacious, anti-biofouling effect can be performed using a single defined species and strain, such as a strain of *S. mutans*, as discussed in multiple studies reviewed herein. Although we identified the most virulent and pathogenic determinant species associated with dental caries, endodontic, and periodontal related pathologies, the interaction between the species in the oral milieu is an important determining factor. Therefore, it is essential to undertake tests evaluating multispecies biofilms. These testing methods may have clinical relevance depending on the type of the model selected, such as defined multispecies biofilm, undefined species (saliva or plaque biofilm) [57,89], or biofilm collected from intraoral appliances [107]. This discussion has been elaborated with details on designing studies with biomaterial interaction in the oral cavity in a recently published perspective, and we encourage readers to refer to this paper [6].

“Zwitterionization” of the biomaterial is seen at the cusp of advancements in biomaterial research with multiple, simultaneous, research paths taking place to improve clinical adaptation. Commonly-used dental materials surface-functionalized with ZPs show promise in combating biofilm growth with exceptional biocompatibility.

The development of newer monomeric units in combination with incorporated bioactive materials and collaborative efforts to assess the bio-interactive nature of biofilm at the interface of dental materials will propel future research in dental materials in a holistic manner.

## Figures and Tables

**Figure 1 ijms-21-09087-f001:**
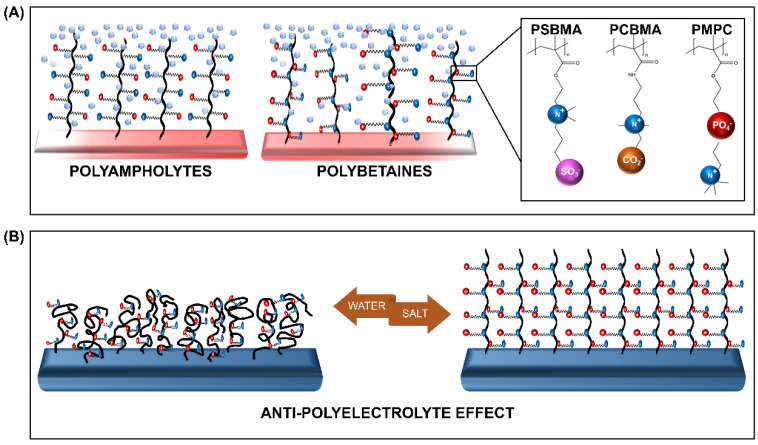
Schematic representation of the (**A**) zwitterionic polymers (ZPs) and (**B**) the salt-responsive antipolyelectrolyte effect (APE) of the ZP brushes. PSBMA—poly (sulfobetaine methacrylate), PCBMA—poly (carboxybetaine methacrylate), and PMPC—poly (2-methacryloyloxyethyl phosphorylcholine).

**Figure 2 ijms-21-09087-f002:**
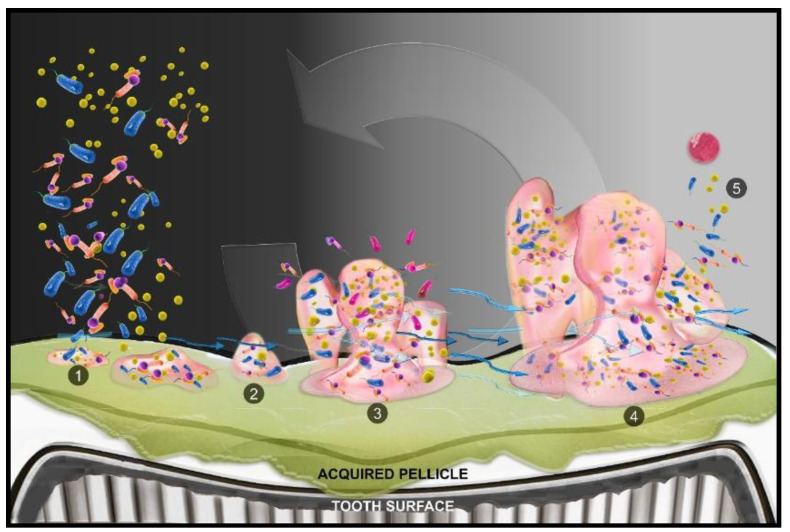
Cycle of biofilm development on the tooth surface. (**1**) Free swimming bacterial cells alight on the tooth surface, conditioned with pellicle derived from salivary proteins, and form clusters. (**2**) Collected cells begin the production of a gooey matrix and cellular signalizing (quorum sensing (QS)) promotes multiplication and colonization. (**3**) Growth of biofilm leads to the development of chemical and oxygen gradients, accompanied by new cell addition to the colony. Water channels develop and pass through the colony. (**4**) Cells and extracellular polymer substance (EPS) detach from the existing colony. (5) Free swimming cells rejoin the microbiome or reinitiate the same process, and expansion of growth of the biofilm occurs.

**Figure 3 ijms-21-09087-f003:**
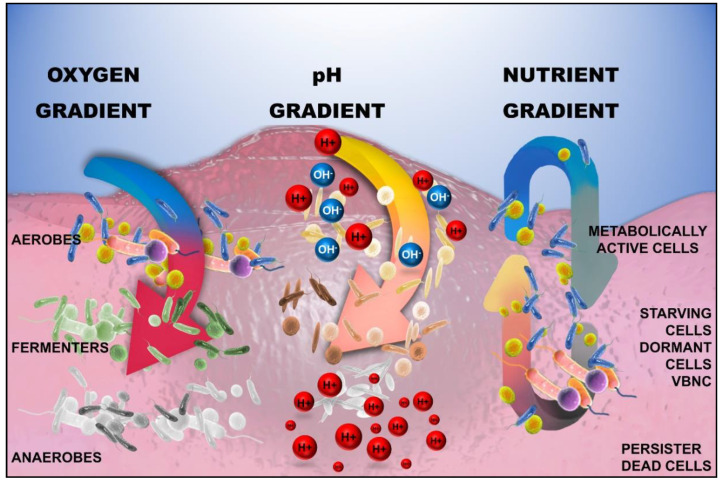
Heterogeneity in the structure of the biofilm due to the presence of oxygen, pH, and nutrient gradients. The utilization of nutrients and oxygen occurs at a rapid rate, resulting in early depletion near the surface, subsequently altering the biochemical milieu with a reduction in pH closer to the attachment surface: H+—Hydrogen ion, OH^−^—hydroxyl ion; VBNC—viable but not culturable cells.

**Figure 4 ijms-21-09087-f004:**
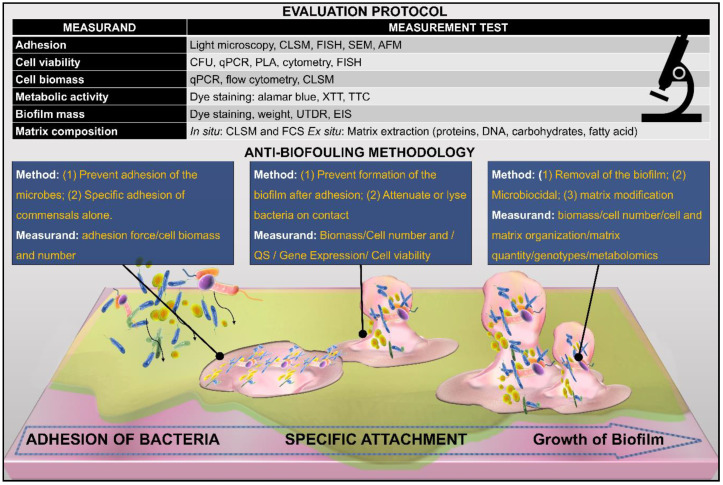
Strategies to combat biofilm development on the surface. Intervention in the material properties can be made to draw an effect at various levels such that it can prevent adhesion and maturation or denatures and kills the constituent microorganism of the biofilm. Each intervention can be objectively analyzed by the corresponding measurands. A detailed analysis of a group of measurands provides a holistic view of the biofilm-resistant mechanism by factoring the different characteristics of the biofilm. See reference [35] for a detailed discussion on measurands and their relation to biofilm characteristics.

**Figure 5 ijms-21-09087-f005:**
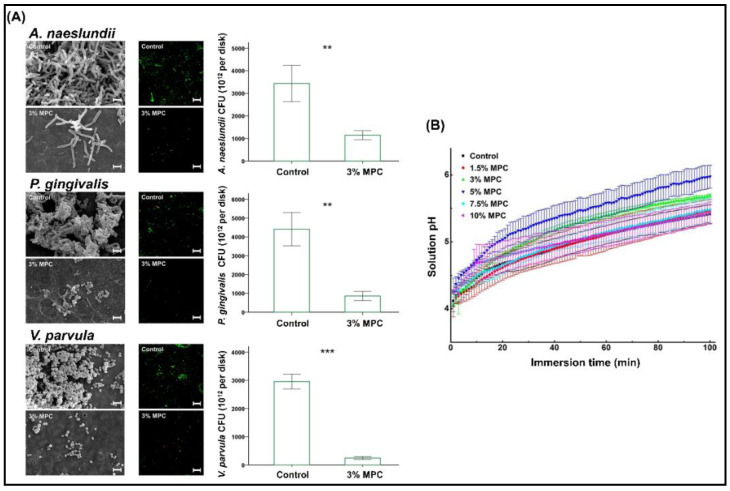
(**A**) Enhanced resistance to bacterial adherence against early and late colonizers with 3% MPC and (**B**) trend in the pH changes observed with different percentages of MPC. MPC—2-methacryloyloxyethyl phosphorylcholine. ** *p* < 0.01, *** *p* < 0.001. Reproduced with permission from Lee et al. [52]; published by Elsevier, 2019.

**Figure 6 ijms-21-09087-f006:**
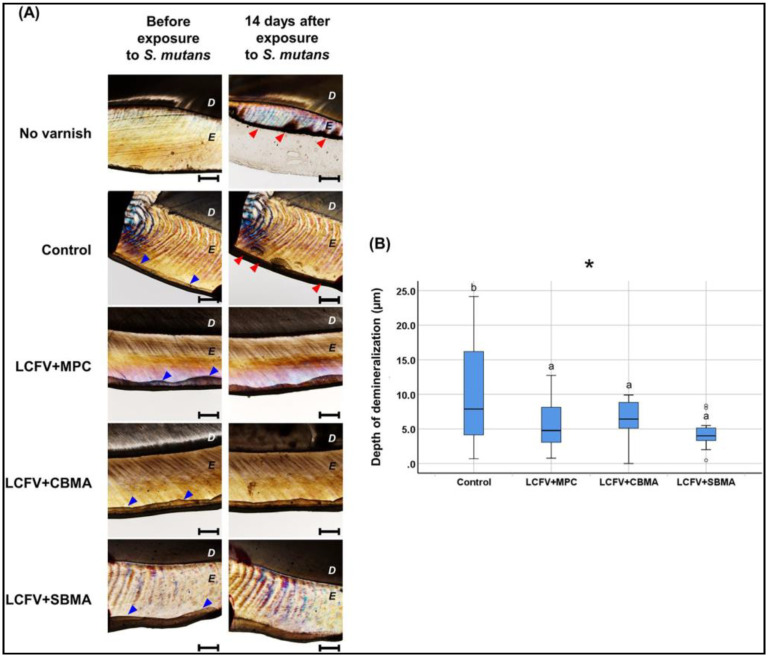
Protective action against demineralization of different polybetaine ZP additions into the fluoride varnish. (**A**) Polarized light microscopic examination and (**B**) the corresponding comparison of the depth of demineralization. Difference in the lower case letters indicate a significant difference between the groups. * *p* < 0.05 [55].

**Figure 7 ijms-21-09087-f007:**
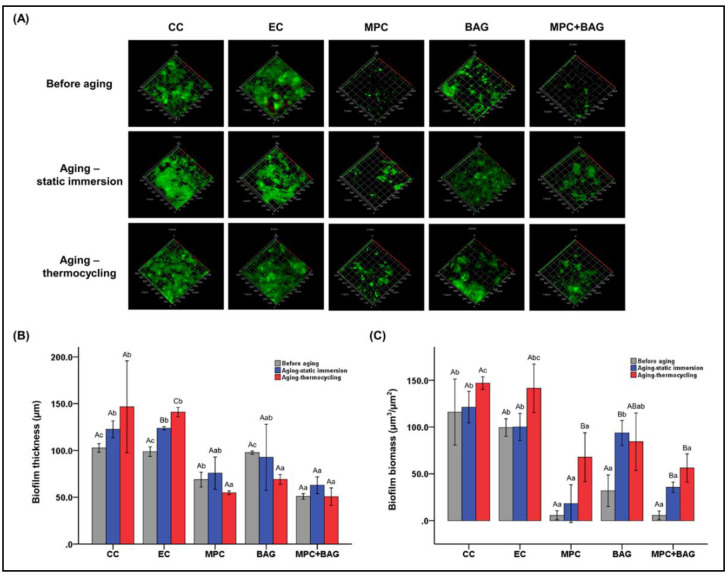
Durable action of MPC added at 3 wt.%, with and without the addition of the bioactive glass (BAG) fillers. (**A**) Confocal laser scanning microscopic images showing extent of the biofilm; (**B**) measurement of the biofilm thickness; and (**C**) biomass at a µm scale. The uppercase and lowercase letters indicate the statistically significant differences within the groups and between groups, respectively at *p* = 0.001 [57].

**Figure 8 ijms-21-09087-f008:**
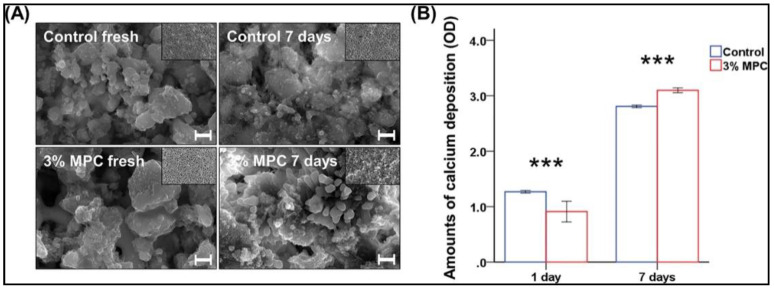
Mineralization ability of the calcium deposition for a one-week interval. Comparison with (**A**) field emission scanning electron microscopy images and (**B**) the optical density (OD) of the amounts of calcium deposition at 3 % MPC. *** *p* < 0.001 for the comparison between control and MPC [76].

**Figure 9 ijms-21-09087-f009:**
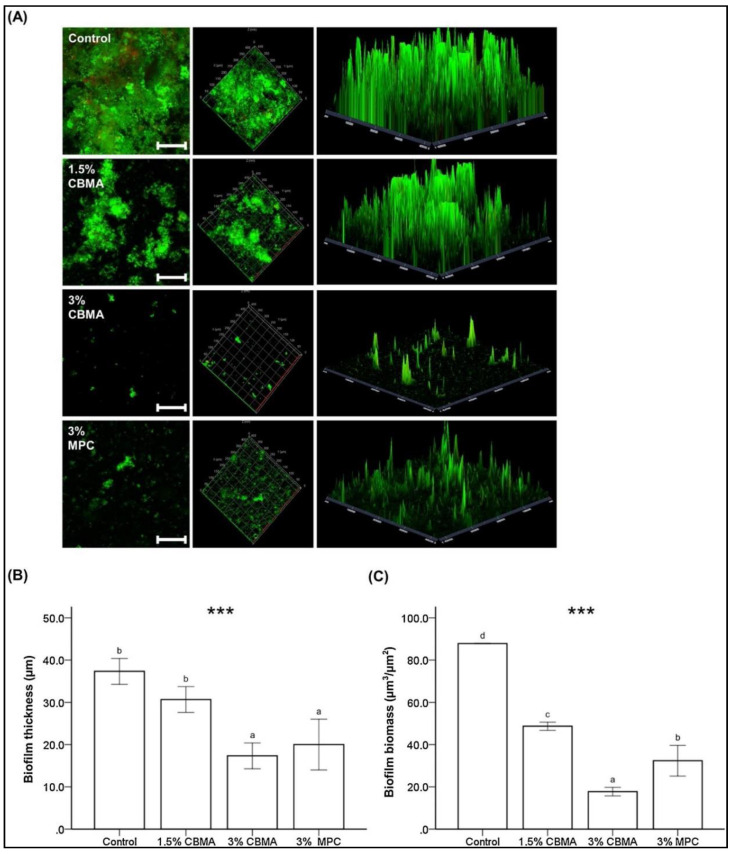
Biofilm resistance of the CBMA-incorporated, poly (methyl methacrylate) (PMMA)-based polymers at different concentrations and their comparison with MPC. (**A**) Confocal laser scanning microscopic images showing the extent of the biofilm; (**B**) measurement of the biofilm thickness; and (**C**) biomass at µm scale. MPC—2-methacryloyloxyethyl phosphorylcholine, CBMA—carboxybetaine methacrylate. Reproduced with permission from Jie et al. Difference in the lower case letters indicate a significant difference between the groups. *** *p* < 0.001 [89]; published by Elsevier, 2020.

**Figure 10 ijms-21-09087-f010:**
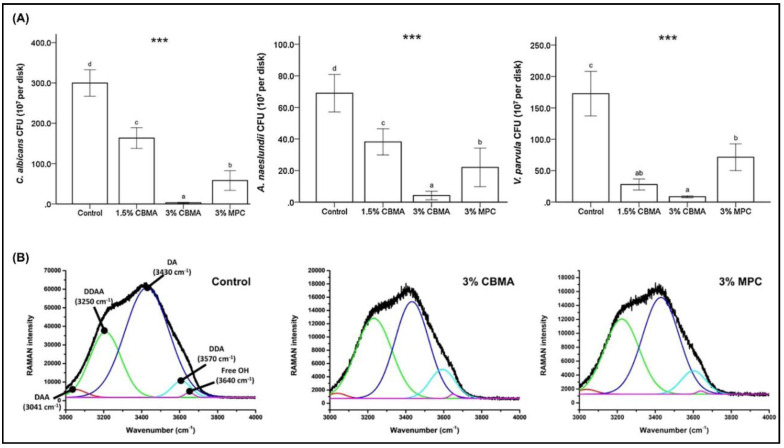
(**A**) Resistance to adherence against fungi and bacteria with addition of CBMA, showing improved performance compared to both the control group and MPC at 3 wt.%. (**B**) Collective band Raman spectroscopy observed for 3% CBMA and 3% MPC. MPC—2-methacryloyloxyethyl phosphorylcholine, CBMA—carboxybetaine methacrylate. The difference in the lower case letters indicate a significant difference between the groups. *** *p* < 0.001. Reproduced with permission from Jie et al. [89]; published by Elsevier, 2020.

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
