# Peer review of "Bio-Interactive Zwitterionic Dental Biomaterials for Improving Biofilm Resistance: Characteristics and Applications"

_ijms, 2020, doi:10.3390/ijms21239087_

Round 1
Reviewer 1 Report
Dear colleagues,
many thanks for the interesting article.
Best wishes
Reviewer 2 Report
Dear authors,
I really appreciated reading your manuscript.
I have only few remarks.
Figures: In some figures (f.e. 6,7, 9) "copied" from the cited paper it is necessary to extend the legend. There are for example in Figure 7 capital letters indicating whether within group comparisons are significant and small letters indicating whether there are significant differences between the materials.
It would be very helpful for the reader to know which letters indicate what. This applies to all figures.
Although many papers have already been mentioned, I believe that the following references would at least partially enrich the article.
High Potential of Bacterial Adhesion on Block Bone Graft Materials
Themistoklis Nisyrios 1 , Lamprini Karygianni 2 , Tobias Fretwurst 1 , Katja Nelson 1 , Elmar Hellwig 3 , Rainer Schmelzeisen 1 , Ali Al-Ahmad 3
Affiliations
• PMID: 32370084
• PMCID: PMC7254222
• DOI: 10.3390/ma13092102
Bacterial adhesion and biofilm formation on yttria-stabilized, tetragonal zirconia and titanium oral implant materials with low surface roughness - an in situ study
Ali Al-Ahmad 1 , Lamprini Karygianni 1 , Max Schulze Wartenhorst 1 , Maria Bächle 2 , Elmar Hellwig 1 , Marie Follo 3 , Kirstin Vach 4 , Jung-Suk Han 5
Affiliations
• PMID: 27093630
• DOI: 10.1099/jmm.0.000267
Microbial adhesion on novel yttria-stabilized tetragonal zirconia (Y-TZP) implant surfaces with nitrogen-doped hydrogenated amorphous carbon (a-C:H:N) coatings
Stefanie Schienle 1 , Ali Al-Ahmad 1 , Ralf Joachim Kohal 2 , Falk Bernsmann 3 , Erik Adolfsson 4 , Laura Montanaro 5 , Paola Palmero 5 , Tobias Fürderer 6 , Jérôme Chevalier 7 , Elmar Hellwig 1 , Lamprini Karygianni 8
Affiliations
• PMID: 26612401
• DOI: 10.1007/s00784-015-1655-5
. 2013 Dec 4;6(12):5659-5674.
doi: 10.3390/ma6125659.
Initial Bacterial Adhesion on Different Yttria-Stabilized Tetragonal Zirconia Implant Surfaces in Vitro
Lamprini Karygianni 1 , Andrea Jähnig 2 , Stefanie Schienle 3 , Falk Bernsmann 4 , Erik Adolfsson 5 , Ralf J Kohal 6 , Jérôme Chevalier 7 , Elmar Hellwig 8 , Ali Al-Ahmad 9
Affiliations
• PMID: 28788415
• PMCID: PMC5452733
• DOI: 10.3390/ma6125659
. 2013 Sep;58(9):1139-47.
doi: 10.1016/j.archoralbio.2013.04.011. Epub 2013 May 18.
In vivo study of the initial bacterial adhesion on different implant materials
A Al-Ahmad 1 , M Wiedmann-Al-Ahmad, A Fackler, M Follo, E Hellwig, M Bächle, C Hannig, J-S Han, M Wolkewitz, R Kohal
Affiliations
• PMID: 23694907
• DOI: 10.1016/j.archoralbio.2013.04.011
